# Treatment Approaches for Oligoprogressive Non-Small Cell Lung Cancer: A Review of Ablative Radiotherapy

**DOI:** 10.3390/cancers17071233

**Published:** 2025-04-05

**Authors:** William Gombrich, Nicholas Eustace, Yufei Liu, Ramya Muddasani, Adam Rock, Ravi Salgia, Terence Williams, Jyoti Malhotra, Percy Lee, Arya Amini

**Affiliations:** 1Kaiser Permanente Bernard J Tyson School of Medicine (KPSOM), Los Angeles, CA 91101, USA; 2Department of Radiation Oncology, City of Hope National Medical Center, Duarte, CA 91010, USA; neustace@coh.org (N.E.);; 3Department of Medical Oncology, City of Hope National Medical Center, Duarte, CA 91010, USA; 4Department of Medical Oncology, City of Hope National Medical Center, Irvine, CA 92618, USA; jymalhotra@coh.org; 5Department of Radiation Oncology, City of Hope National Medical Center, Irvine, CA 92618, USA; percylee@coh.org

**Keywords:** non-small cell lung cancer (NSCLC), radiation, stereotactic body radiation therapy (SBRT), stereotactic ablative radiotherapy (SABR), oligoprogression, oligometastasis

## Abstract

Radiation therapy historically played a more palliative role in the setting of metastatic lung cancer. However, more recent data appears to suggest a potential benefit for more aggressive local ablative therapy (LAT) with stereotactic body radiation therapy (SBRT), also referred to as stereotactic ablative radiotherapy (SABR). In patients with oligoprogressive disease (defined typically as 5 or less sites of progression), the theoretical purpose of SBRT to sites of progression is to help maintain patients on their current therapy and prevent/delay the need to switch to next-line systemic therapy. Here we review the current literature evaluating the role of SBRT in oligoprogressive lung cancer and future directions.

## 1. Introduction

It is estimated that there are 234,580 cases of lung cancer diagnosis and 125,070 deaths due to lung cancer in the United States in 2024 [1]. Of these cases, it is proposed that approximately 80% of these cases will be due to NSCLC, which makes identifying a comprehensive and effective treatment to increase overall survival (OS) and PFS important. Hellman and Weichselbaum [2] coined the term oligometastatic as an intermediate state of cancer spread characterized by restricted metastatic burden. This unique population is thought to be one in which LAT, such as radiation or surgery, could prolong survival or potentially be curative in some patients. A similar principle has been applied in the oligoprogressive setting, which is defined as those progressing in a limited number of sites, often defined as 3–5 sites of progression, where potentially local therapy can either delay the need to resume systemic therapy or switch to the next line of therapy. In 2020, the European Society for Therapeutic Radiology and Oncology (ESTRO) and the European Organization for Research and Treatment of Cancer (EORTC) issued a consensus and decision tree for classifying oligometastatic disease. They defined oligometastatic disease as the intermediate state between localized cancer and widespread systemic metastasis [3]. The classification distinguishes between two broad categories: de-novo oligometastatic disease and induced oligometastatic disease.

Genuine oligometastatic disease refers to patients with no prior polymetastatic disease suggesting lower metastatic potential. It is further subdivided into de-novo oligometastatic disease (no prior oligometastatic diagnosis), which can present as synchronous (identified at the same time as the primary tumor), metachronous oligorecurrence (recurrence after a disease-free interval), or metachronous oligoprogression (progression in a limited number of metastatic sites); or present as repeat oligometastatic disease (previously diagnosed oligometastatic disease), further classified as repeat oligopersistent disease (persistent oligometastases despite treatment) or repeat oligoprogressive disease (progression in a limited number of metastatic sites after prior oligometastatic treatment). Induced oligometastatic disease occurs in patients with a history of polymetastatic disease who have responded well to systemic therapy, leaving only a few residual sites of disease. This category, associated with a higher metastatic potential, is further classified into induced oligorecurrence (recurrence in a limited number of sites after systemic treatment), induced oligopersistence (persistent but controlled oligometastases following therapy), or induced oligoprogression (progression in a few remaining metastatic sites despite treatment). This classification framework provides a structured approach to understanding oligometastatic disease, aiding in prognosis assessment and treatment planning [3].

Much work continues to be done to reach a consensus on what defines an oligometastatic state and which patients may benefit from a combined local and systemic treatment plan with a potentially curative intent. In this review, we explore the evolving landscape of oligoprogressive NSCLC, which is typically defined as limited metastatic sites that have continued to progress despite initial systemic or targeted therapy [3,4]. We discuss the evolving landscape of oligoprogressive NSCLC treatment, summarizing evidence from numerous studies and discussing possible future directions in the management of this disease state.

## 2. Methods

A systematic review of the current medical literature from peer-reviewed journals was conducted from 1 January 2010 to 1 December 2024. The search strategy was developed based on National Library of Medicine^®^ Medical Subject Headings (MeSH^®^) with the addition of subject-specific keywords (https://www.ncbi.nlm.nih.gov/mesh, accessed on 20 March 2025). The bibliographies of full articles were reviewed to include studies which were potentially relevant. The literature was reviewed for the quality of study design, cohort size, selection bias, evaluation of participants in relation to time from exposure, and methods of assessments. Search criteria included the following: all adults (18 plus years), prospective and retrospective studies, case series, review papers, and meta-analyses. Case reports, letters, editorials, abstracts, and corrections/errata were excluded.

## 3. Historical Context and Evolution of Treatment Approaches

The evolution of treatment for oligoprogressive NSCLC has been marked by advances in targeted therapies, immunotherapy, and local ablative techniques, such as stereotactic body radiation therapy/stereotactic ablative radiation therapy (SBRT/SABR). Since the mid-1990s, research has increasingly focused on combining systemic and local treatments to optimize outcomes. The concept of oligoprogression gained traction in the early 2010s, with studies beginning to assess whether targeted local interventions could improve outcomes for patients with limited progression while on systemic therapies. Presently, larger prospective trials are limited in the oligoprogressive setting. In 2012, Weickhardt et al. conducted a retrospective study evaluating the effects of LAT on oligoprogressive patients with metastatic NSCLC harboring epidermal growth factor receptor (EGFR) mutations or anaplastic lymphoma kinase (ALK) rearrangements. The study was among the first to suggest that LAT, when combined with continuous tyrosine kinase inhibitor (TKI) treatment, could prolong disease control even after initial systemic progression. The results indicated that patients treated with LAT had a median PFS of 9.0 months for ALK-positive cases and 13.8 months for EGFR-mutant cases, underscoring the potential role of LAT in extending the benefits of targeted therapy [5]. Two years later, in 2014, Iyengar et al. conducted a phase 2 trial evaluating SBRT combined with erlotinib in patients with oligoprogressive disease who progressed on chemotherapy, demonstrating that combining SBRT with erlotinib in patients with oligoprogressive NSCLC not only extended PFS but also improved local control within the SBRT-treated fields, suggesting that local therapies could complement systemic agents to target resistant cancer clones [6]. This, along with another 2014 retrospective observational cohort study investigating the use of hypo-fractionated high-dose radiotherapy (HDRT) in 36 patients with EGFR-mutant NSCLC experiencing progression while on gefitinib and found HDRT prolonged disease control, and laid the foundation for researching possible treatment plans for oligoprogressive NSCLC [7].

Since then, other observational retrospective and prospective studies, as well as phase 2 trials, have been conducted, further contributing to evidence suggesting local ablative therapy improves morbidity and mortality in patients with oligoprogressive NSCLC (Table 1).

## 4. SBRT in Oligoprogressive Oncogenic-Driver Positive

Targeted therapies have become a cornerstone in the management of NSCLC, particularly for patients with driver mutations such as EGFR or ALK rearrangements. As mentioned earlier, Weickhardt et al. was one of the first to suggest that continuing targeted therapies beyond progression, combined with LAT, could benefit patients with oncogene-driver mutated NSCLC [5]. Stereotactic body radiation therapy (SBRT) was used for extra-CNS lesions with doses ranging from 15–54 Gy (median: 40 Gy) and stereotactic radiosurgery (SRS) was used for central nervous system (CNS) lesions. The results indicated that LAT could delay the need for switching to more aggressive systemic therapies, especially in patients with CNS involvement. In a 2014 study by Gan et al., researchers conducted a retrospective analysis of ALK-rearranged patients receiving crizotinib and found that LAT was associated with durable local control. LAT included SBRT (dose range: 12–54 Gy in 1–3 fractions) and hypofractionated radiotherapy (dose range: 30–40 Gy in 10 fractions). The 12-month actuarial local control (LC) rate was 86%, indicating LAT’s effectiveness in controlling progressive lesions. Additionally, LAT allowed patients to continue TKI therapy beyond progression, which was associated with prolonged PFS. This ability to maintain patients on their current targeted therapy regimen while effectively managing disease progression highlights LAT’s role in the personalized management of ALK-rearranged NSCLC [25].

More recently, a 2024 retrospective study involving 73 patients with ALK-rearranged NSCLC reported that patients who continued TKI therapy beyond oligoprogression, especially when combined with LAT, experienced a significant clinical benefit. The study emphasized the value of molecular profiling to guide treatment decisions and tailor LAT to specific progression patterns, such as CNS versus extracranial metastases [13].

The role of LAT in EGFR-mutated NSCLC has been similarly transformative. In a 2017 study, Qiu et al. found that patients who underwent LAT had a median OS of 35.0 months, with significant predictors of outcomes including EGFR mutation type and the time from first progression to LAT [24]. Another study conducted in 2019 with 206 EGFR-mutated patients demonstrated that LAT significantly improved PFS, with patients experiencing median PFS1 and PFS2 durations of 10.7 months and 18.3 months, respectively [22]. A second 2019 retrospective study continued to explore this strategy with the third-generation TKI osimertinib, which showed promise in extending the median OS to 28 months and an overall response rate of 80% with a disease-control rate of 92% when combined with LAT for patients experiencing oligoprogression [21]. A recent Swiss multicenter retrospective study of EGFR-mutant NSCLC patients treated with first-line osimertinib found that 77% of patients experienced oligoprogressive disease, with the lung (62%) and brain (30%) being the most common progression sites. Patients with oligoprogressive disease had a significantly improved overall survival compared to those with systemic progression (51.6 vs. 26.4 months, *p* = 0.004). Importantly, 45% of oligoprogressive disease patients received LAT (radiotherapy or surgery), which was associated with the longest OS of 60 months—suggesting that LAT may help delay the need for systemic therapy switches and prolong TKI efficacy. Despite the well-documented CNS penetration of osimertinib, 30% of patients still developed brain progression, highlighting the importance of routine magnetic-resonance imaging (MRI)-based surveillance [11].

The integration of LAT with targeted therapies represents a paradigm shift in the management of oligoprogressive NSCLC. By providing durable local control and delaying systemic therapy transitions, LAT offers a personalized approach to addressing disease progression. As molecular profiling and patient selection continue to refine treatment strategies, LAT is poised to play an increasingly integral role in the care of oligoprogressive NSCLC patients.

## 5. SBRT in Oligoprogressive Oncogenic-Driver Negative

The creation of immune checkpoint inhibitors (ICIs) has put forth new opportunities for combining local ablative therapies with immunotherapy in managing patients with oligoprogressive NSCLC. Commonly used ICIs include target programmed death-1 (PD-1), programmed death-ligand 1 (PD-L1), and cytotoxic T-lymphocyte-associated protein 4 (CTLA-4). PD-1 and PD-L1 inhibitors, including pembrolizumab, nivolumab, and atezolizumab, are widely used, especially in patients with a high PD-L1 expression or in combination with chemotherapy for broader patient populations. Patients receiving ICI treatments are still capable of developing resistance to these checkpoint inhibitors via specific tumor subclones. Managing oligoprogression in this setting includes radiation and surgical approaches which can help eliminate these resistant subclones while maintaining systemic control. Continuing ICIs despite oligoprogression may be beneficial; however, the emergence of new resistance mechanisms may require a transition to alternative systemic therapies.

Rheinheimer et al. (2020) reported that oligoprogressive disease occurred later and affected fewer sites in patients treated with first-line immunotherapy monotherapy compared to chemoimmunotherapy [20]. The study also found that patients with oligoprogression had a higher PD-L1 expression, which was associated with improved outcomes. Building on these findings, a 2021 retrospective study conducted at a single center analyzed 24 patients with advanced NSCLC who developed oligoprogression following an initial response to ICI’s and subsequently received SBRT. The median post-oligoprogression progression-free survival was 11 months, and the median overall survival after oligoprogression was 34 months. Patients with adenocarcinoma, a higher lung immune prognostic index, and positive PD-L1 expression, were more likely to achieve favorable survival outcomes. Specifically, SBRT demonstrated a high local control rate, with a median local control duration of 43 months and a 2-year local control rate of 81.8%. The study suggests that SBRT may be a promising strategy to delay systemic therapy and extend the benefits of ongoing ICI treatment in patients with oligoprogressive disease while maintaining an acceptable safety profile [19].

Similarly, a 2022 retrospective cohort study evaluated the use of SBRT in patients with oligoprogression while on ICIs. The study demonstrated that SBRT was associated with an improved local control and prolonged PFS, especially in patients with a higher PD-L1 expression and a longer duration of ICI treatment before oligoprogression [17]. These findings support the integration of LAT with immunotherapy as a strategy to extend the benefits of ICIs. Furthermore, a 2024 prospective multicenter observational study involving SBRT combined with anti-PD-1 therapy in NSCLC and melanoma patients showed a positive and durable response in patients receiving combination SBRT in the setting of progression, with a median PFS of 14.2 months [16]. Of note, approximately 64% of patients experienced an abscopal effect where 1–2 predefined nonirradiated lesions had a ≥30% response. The abscopal effect, where localized radiation induces a systemic immune response, highlights the potential synergistic effects of combining radiotherapy with ICIs, and further studies are needed to better understand this effect and potentially how to reproduce it more frequently in our patients.

A 2024 study on failure patterns in metastatic NSCLC treated with first-line pembrolizumab showed that oligoprogression occurred in 39.9% (*n* = 79) of patients. This group exhibited a longer OS compared to those with poly-progression. Patterns of failure were categorized as local, regional, distant, or any combination. Failures were also analyzed based on whether the progression occurred in existing lesions, new lesions, or both. Most failures were distant (43.9%, *n* = 87) or a combination of locoregional and distant sites (34.4%, *n* = 68). Failure in existing lesions alone was observed in 33.3% (*n* = 66) of cases, while 45.0% (*n* = 89) showed a combination of new and existing lesions. Patients with progression in existing lesions alone had an improved OS (median 28.7 months) versus those with new lesions or both new and existing lesions (median 13.9 months). Patients with oligoprogression treated with RT to all of the sites of progression had a median OS of 62.2 months compared to 22.9 months for those who switched systemic therapies [15].

The combination of ICIs and LAT represents a promising strategy for managing oligoprogressive NSCLC, offering both local and systemic disease control. Studies consistently show that LAT can prolong the benefits of ICIs, particularly in patients with a higher PD-L1 expression and limited disease progression. As understanding of the synergistic approach broadens, it may pave the way for more personalized and effective treatment strategies.

## 6. SBRT in Oligoprogressive Oncogenic-Driver Positive and Negative Patients

The benefits of LAT are not confined to NSCLC with specific driver mutations. A 2021 retrospective, multicenter cohort study found that metastasis-directed stereotactic radiosurgery (SRS) combined with targeted therapy or immunotherapy improved the OS and PFS in patients with oligoprogressive disease compared to those with poly-progressive disease. Patients had concurrent targeted therapy and immunotherapy, which included EGFR/ALK inhibitors (60%) and immune checkpoint inhibitors (31%). The study evaluated both cranial and extracranial SBRT/SRS, with brain metastases as the most frequently treated location (68%) [18]. The findings demonstrated that patients with fewer affected organs and those in earlier lines of therapy experienced better outcomes. Additional studies focusing on SBRT in oligoprogressive NSCLC patients on immunotherapy are discussed in the following section.

Expanding on this, a 2023 study involving 168 patients treated with SBRT for oligoprogressive or oligorecurrent NSCLC found that LAT significantly extended PFS, particularly in patients with fewer metastatic lesions and a lower cumulative tumor burden [14]. High programmed death-ligand 1 (PD-L1) expression and EGFR alterations were associated with worse PFS, while ALK rearrangement was associated with a better OS and PFS in univariate analysis. Despite this, the benefit was not seen in multivariate analysis, potentially underscoring the importance of patient selection and tumor biology, as LAT may be more beneficial for individuals with limited disease progression.

In a study by Chou et al., 23 patients with oligoprogressive NSCLC on maintenance therapy received radiotherapy to all progressive sites, achieving a high local control rate of 97.5%. Radiotherapy was delivered by SBRT (50–60 Gy in 3–5 fractions) or hypofractionated radiation (45–60 Gy in 15 fractions). Patients without prior radiation experienced a significantly longer PFS (11.9 months vs. 6.2 months, *p* = 0.018), supporting the use of LAT to delay disease progression and extend maintenance therapy duration [8].

## 7. Challenges and Future Directions in Oligoprogressive Disease

The 2024 STOP trial, a phase 2, multicenter, open-label, randomized controlled trial, offered insights into LAT’s efficacy in various cancers. Ninety patients with 1–5 oligoprogressive metastases were randomized to standard systemic therapy alone or systemic therapy plus SBRT. The trial showed SBRT provided superior lesional control but did not extend PFS or OS. Challenges included slow patient accrual, high dropout rates in the SOC arm, and protocol deviations (e.g., some SOC patients received ablative treatments). The authors hypothesized that SBRT may benefit selected patient subgroups (e.g., those with specific histologies or fewer metastases), advocating for more focused trials to reduce patient heterogeneity. Challenges such as high crossover rates and cancer-type heterogeneity in this trial highlighted the need for more directed studies on LAT’s efficacy in specific cancer subgroups [10].

The Memorial Sloan Kettering Cancer Center (MSKCC) CURB trial demonstrated that adding SBRT to standard-of-care systemic therapy significantly improved median PFS in NSCLC patients (10.0 months vs. 2.2 months for the control group). Patients were randomly assigned (1:1) to either standard of care systemic therapy or standard of care combined with SBRT. The most common SBRT regimen was 27–30 Gy in three fractions and 30–50 Gy in five fractions. Stratification was applied based on factors including the number of progressive sites, receptor/genetic alteration status, primary cancer type, and previous systemic therapies. While SBRT did not extend the OS, it did prolong the time patients remained on their current systemic therapy, with NSCLC patients showing a median duration of 11.0 months on standard of care plus SBRT compared to only 3.9 months in breast cancer patients [12]. These phase 2 findings are the basis of a planned cooperative group phase 3 trial referred to as CURB2, evaluating the role of SBRT in oligoprogressive NSCLC on an ICI.

As the field continues to evaluate the role of LATs in the setting of oligoprogressive disease, better risk stratification is needed to define those who truly have oligoprogressive disease and may benefit from LAT compared to those who may harbor micrometastatic disease not evident on imaging, who benefit primarily from changing systemic therapy. Detecting circulating tumor DNA (ctDNA) is one potential way to stratify these patients likely to benefit from LAT that ongoing studies are evaluating. Currently, there are several retrospective studies suggesting ctDNA may function as a useful biomarker to identify the oligometastatic and oligoprogressive patients who may benefit the most from LATs [26,27].

In addition, a better attention to patterns of failure (e.g., liver, CNS, lungs, etc.), quality of life, the cost-effectiveness of LAT, measuring toxicities from different forms of LAT (e.g., SBRT, needle ablation, etc.), and developing a consensus on which efficacy, toxicity, and QOL endpoints to measure will be critical to moving the field forward. For example, in order to better assess the effects of LAT on halting disease progression, studies should include a “time to development of new metastases” after LAT rather than measuring only PFS (since PFS includes progression at any point including the ablated lesions).

## 8. Study Types and Evidence Quality

The literature on oligoprogressive NSCLC encompasses a variety of study designs, including retrospective reviews, prospective trials, and phase 2 clinical studies, with most of the studies being observational. Prospective blinded randomized control trials are inherently difficult to perform in this domain given the nature of treatment (radiation, systemic/targeted therapy side-effects, crossover, etc.), the lack of control arms, and the heterogeneity of the population concerning disease biology, demographics, and treatment course. Therefore, retrospective analyses offer insights into real-world practice, often focusing on the outcomes of patients treated with LAT and targeted or systemic therapy but are intrinsically at risk of bias and outside confounding factors that cannot be accounted for [5,21,22]. Prospective studies and clinical trials provide higher-quality evidence, examining the efficacy of novel therapeutic combinations, including SBRT combined with targeted therapies or ICIs, but feasibility has prevented studies from adequate patient accrual, follow-up, and randomization due to the reasons previously mentioned [6,12,23]. Consequently, a call for a novel approach to conduct large multicenter randomized control trials has been made by many [4,9,10,12,13,16,24].

## 9. Conclusions

The landscape of oligoprogressive NSCLC management has evolved in recent years with the integration of LAT and systemic treatments that offer new promises to patients in whom disease progression is limited. Currently, to those of whom LAT has shown considerable benefit when used in combination with targeted therapies for EGFR and ALK alterations, LAT potentially offers prolonged PFS, delayed transitions to systemic therapy, and an improved local control when used at sites of disease progression where therapeutic resistance has developed. LAT is especially beneficial in the setting of CNS progression for patients’ driver mutations and underscores the importance of multimodality treatment approaches.

Similarly, combining LAT with ICIs opens new perspectives in the treatment of oligoprogressive disease. Growing evidence underlines that LAT may prolong the benefit from ICIs and much work is ongoing to use LAT to elicit systemic immune responses that is outside the scope of this review. Currently, the data suggests that those with a high PD-L1 expression and fewer sites of progression are most likely to benefit, which suggests biomarker-driven approaches are critical in optimizing treatment outcomes.

Despite these advances, challenges persist in interoperating current data, including heterogeneity in study designs, patient populations, and limited phase 3 clinical trial data. The continued integration of molecular profiling and biomarkers, such as ctDNA, into future studies is the key to defining patient selection and risk stratification for those that benefit from LAT. Addressing these gaps, future studies will refine the role of LAT in combination with systemic therapies and drive more effective and personalized treatment strategies for oligoprogressive NSCLC, leading to an improved survival and quality of life in this unique patient population.

## Figures and Tables

**Table 1 cancers-17-01233-t001:** Study designs and outcomes among studies examining oligoprogressive NSCLC.

Author(s), Year	Study Design	Population	Intervention/Comparison	Outcomes	Key Findings	Limitations
Chou et al., 2024 [8]	Retrospective analysis	23 NSCLC patients with 1–5 progressive sites	Radiotherapy to all oligoprogressive lesions + systemic therapy (TKI, ICI, and chemotherapy)	Median OS: 31.3 months; median PFS: 8.4 months	RT yielded high local control (97.5%); improved PFS in those without prior radiation	Small cohort; retrospective single-institution study
Glicksman et al., 2024 [9]	Phase 2 single arm trial	70 patients with GU, breast, or GI cancers	SBRT + ongoing systemic therapy	1-year PFS: 32%, OS: 75%, low toxicity rates	SBRT delayed systemic therapy changes in 47% of patients; low-grade toxicity	Single-center study, limited generalizability
Schellenberg et al., 2024 [10]	Randomized Phase 2 Trial	90 patients with ≤5 progressive lesions on systemic therapy	SBRT + SOC vs. SOC alone	No significant difference in PFS (8.4 vs. 4.3 months) or OS (31.2 vs. 27.4 months); lesional control improved (70% vs. 38%)	SBRT improved local control without increasing grade 4/5 toxicity; OS/PFS benefits inconclusive	High dropout rate in SOC arm; heterogeneity in histologies
Schuler et al., 2024 [11]	Retrospective analysis	148 EGFR-mutated NSCLC patients	First-line osimertinib + LAT	OS: 51.6 months; LAT group OS: 60 months	LAT extended OS in patients with OPD; most progression in lungs and CNS	Retrospective design; variability in LAT usage
Tsai et al., 2024 [12]	Phase 2 RCT	106 NSCLC or breast cancer patients	SBRT + SOC vs. SOC alone	PFS: 10 months (SBRT) vs. 2.2 months (SOC)	SBRT improved PFS significantly in NSCLC but not in breast cancer	Premature closure due to accrual challenges
Wu et al., 2024 [13]	Retrospective analysis	73 ALK-positive NSCLC patients with progression on ALK-TKIs	Local therapy vs. sequential ALK-TKIs	Local therapy extended targeted therapy by 6.4 months; OS: NA vs. 11.9 months for resistant ALK mutations	Local therapy effective in managing oligo-progression and prolonging ALK-TKI benefits	Single-center; heterogeneous treatments; limited sample size
Ebadi et al., 2023 [14]	Retrospective analysis	168 NSCLC patients with oligoprogressive/oligorecurrent disease	SBRT to 1–5 lesions	Median OS: 31 months; PFS: 6.6 months; TNT-D: 9 months	SBRT safe and effective; patients with ≤2 lesions had better OS and TNT-D than those with 3–5 lesions	Retrospective design; heterogeneous systemic therapies
Friedes et al., 2024 [15]	Retrospective analysis	298 mNSCLC patients on pembrolizumab	Radiation vs. systemic therapy after PD	OS: 62.2 months with radiation vs. 22.9 months	Radiation for OPD improved OS significantly	Retrospective, single-center design
Chicas-Sett et al., 2022 [16]	Prospective observational	50 NSCLC/melanoma patients, oligoprogression on anti-PD-1	SBRT + anti-PD-1 ICI	Median PFS: 14.2 months; median OS: 37.4 months	SBRT combined with anti-PD-1 achieved high response rates, with 65% abscopal responses	Lack of randomization; single-region cohort
Mahmood et al., 2022 [17]	Retrospective analysis	120 solid tumor patients with ≤5 lesions	Radiation to progressive lesions + ongoing ICI	Median OS: 29.8 months; LC at irradiated sites improved outcomes	Favorable outcomes with radiation; PFS and OS linked to PD-L1 expression	Retrospective; heterogeneity in patient population
Kroeze et al., 2021 [18]	Retrospective multicenter	108 stage IV NSCLC patients	MDT with concurrent TT/IT	2-year OS: 51% (OPD) vs. 25% (PPD)	MDT was safe and improved survival; 58% remained on same TT/IT	Retrospective data, heterogeneity in treatments
Wang et al., 2021 [19]	Retrospective analysis	Four NSCLC patients with CPI resistance	SBRT + CPIs	Median OS: 34 months, PFS: 11 months	Combined SBRT and CPI improved LC and survival	Small sample size; lack of control group
Rheinheimer et al., 2020 [20]	Retrospective analysis	372 NSCLC patients on IO (PD-1/PD-L1 inhibitors)	IO monotherapy vs. chemo-IO; LAT for OPD	Median OS: 26 months; median TTP for OPD: 11 months	LAT improved control for OPD (50% treated); IO alone showed less frequent OPD vs. TKI-treated patients	Retrospective; no standardization for imaging intervals
Santarpia et al., 2020 [7]	Retrospective cohort	36 EGFR-mutated NSCLC patients	HDRT + ongoing TKI	OS: 38.7 months, overall PFS: 18.8 months	HDRT + TKI significantly prolonged disease control	Small sample size, single-center real-world data
Schmid et al., 2019 [21]	Retrospective multicenter	50 EGFR T790M+ NSCLC patients on osimertinib	LAT vs. second line systemic therapy for progression	Median OS: 28 months; LAT extended osimertinib treatment duration (19.6 vs. 7 months)	High oligo-PD rate (73%); LAT beneficial with osimertinib continuation	Retrospective; heterogeneous patient population
Xu et al., 2019 [22]	Retrospective study	206 EGFR-mutated NSCLC patients with oligoprogression	EGFR-TKI continuation + LAT	Median OS: 37.4 months; 2-year OS: 78.9%	LAT prolonged OS and reduced progression in EGFR-mutant patients	Single-center; retrospective design
Weiss et al., 2019 [23]	Phase II prospective trial	25 EGFR-mutated NSCLC patients, oligoprogression	SBRT for ≤3 sites + TKI (erlotinib)	Median OS: 29 months; Median PFS: 6 months	SRT extended PFS with manageable toxicity in select patients	Small cohort; no control group
Qiu et al., 2017 [24]	Retrospective cohort	46 EGFR-mutated NSCLC patients	Local therapy + EGFR-TKI	Median OS: 13 months post-LT; PFS: 7 months	LT and TKI continuation feasible; EGFR mutation type affects outcomes	Small sample size; retrospective nature
Gan et al., 2014 [25]	Retrospective analysis	33 ALK-positive NSCLC patients	LAT for oligoprogression + crizotinib	12-month local control rate: 86%; OS: 72% (2 years)	LAT allowed prolonged use of crizotinib with high local control rates	Retrospective; single institution
Iyengar et al., 2014 [6]	Phase II single-arm trial	24 NSCLC patients with ≤6 progressive lesions	SBRT to all lesions + erlotinib	Median OS: 20.4 months; median PFS: 14.7 months	SBRT with erlotinib was well tolerated and improved survival	Single-arm study; unselected population
Weickhardt et al., 2012 [5]	Retrospective study	51 NSCLC patients (ALK/EGFR positive)	LAT + ongoing TKI (crizotinib/erlotinib)	Median PFS2: 6.2 months	LAT extended TKI benefits; suitable for CNS/systemic progression	Small sample; single-institution retrospective study

Table 1 Key: Anaplastic lymphoma kinase tyrosine kinase inhibitor (ALK-TKI), epidermal growth factor receptor (EGFR), local ablative therapy (LAT), local control (LC), non-small cell lung cancer (NSCLC), overall survival (OS), progression-free survival (PFS), stereotactic body radiation therapy (SBRT), standard of care (SOC), metastasis directed therapy (MDT), high-dose-rate therapy (HDRT), immune checkpoint inhibitor (ICI), time to next treatment or death (TNT-D), oligoprogressive disease (OPD), immune checkpoint inhibitor (ICI).

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
