# Peer review of "Treatment Approaches for Oligoprogressive Non-Small Cell Lung Cancer: A Review of Ablative Radiotherapy"

_cancers, 2025, doi:10.3390/cancers17071233_

Round 1
Reviewer 1 Report
Comments and Suggestions for Authors
Overall, this review is kind superficial. In addition, it lacks:
1) how the literature search was made,
2) based on what rationale the references used were chosen, and
3) discussion.
Additionally, it would be great to provide more in-depth information of each study, if available, such as:
1) prescription dose and fraction size,
2) target location (e.g., central or not), and
3) motion management strategy.
Review articles often attract not only experts but also more general readers. Consider providing full spells for every acronym. Having a glossary would be a good approach.
There is no mention about the table in the main text.
Include the reference number for each study in the table. Also indicate which section it does belong to (i.e., oncogenic-driver positive, negative or both).
Minor Comments:
Lines 83,126 and 180; use either one, between "Oncogenic-Driver" or "Oncogenic Driver"
Line 213; a typo "benefit select patient" => "benefit selected patient"
Author Response
Overall, this review is kind superficial. In addition, it lacks:
1) how the literature search was made,
We have added a methods section to describe the process we used to search for relevant papers.
2) based on what rationale the references used were chosen, and
3) discussion.
We have added a methods section to describe the process used to find and identify the papers in this review. Thank you.
Additionally, it would be great to provide more in-depth information of each study, if available, such as:
1) prescription dose and fraction size,
2) target location (e.g., central or not), and
3) motion management strategy.
We have attempted to go back and add some detail based on what is available. The limitation of all these studies is that they generally do not include a range of doses, tumor locations, etc. A number of studies did not include radiation details unfortunately.
Review articles often attract not only experts but also more general readers. Consider providing full spells for every acronym. Having a glossary would be a good approach.
Thank you for the suggestion. We have gone through and made sure the acronyms are spelled out the first time they are being used.
There is no mention about the table in the main text.
Thank you for noting this. We have added in the reference to the table.
Include the reference number for each study in the table. Also indicate which section it does belong to (i.e., oncogenic-driver positive, negative or both).
Thank you for the suggestion. We have added the reference number to each study. In regards to the oncogenic driver status, this is a bit challenging as multiple are not otherwise specified and/or including both. We therefore in column 4 of the the table tried to specify when it was made clear what general therapy agent they received (TKI, IO etc). If additional clarification or information is needed please let us know. Many of these trials are basket trials and include all comers.
Minor Comments:
Lines 83,126 and 180; use either one, between "Oncogenic-Driver" or "Oncogenic Driver"
Corrected thank you
Line 213; a typo "benefit select patient" => "benefit selected patient"
Corrected thank you
Reviewer 2 Report
Comments and Suggestions for Authors
The authors conducted a review of treatment approaches for oligoprogressive NSCLC. There are several subtypes of oligometastases, among which oligoprogressive disease is a particularly hot topic in this field.
The manuscript is clearly written and well-structured. The reviewer has only a minor comment:
Oligoprogressive disease differs from genuine oligometastatic disease, such as de novo oligometastases. Therefore, the authors may wish to explain the biological differences between these two entities and clarify the position of oligoprogression within the spectrum of oligometastatic subgroups.
Author Response
The authors conducted a review of treatment approaches for oligoprogressive NSCLC. There are several subtypes of oligometastases, among which oligoprogressive disease is a particularly hot topic in this field.
The manuscript is clearly written and well-structured. The reviewer has only a minor comment:
Oligoprogressive disease differs from genuine oligometastatic disease, such as de novo oligometastases. Therefore, the authors may wish to explain the biological differences between these two entities and clarify the position of oligoprogression within the spectrum of oligometastatic subgroups.
Thank you for the feedback, we have clarified the classification of Oligometastasis and its subtypes in the introduction.
Reviewer 3 Report
Comments and Suggestions for Authors
Major points:
I would like to congratulate the authors for a timely review of a hot oncologic topic in oncology. The article's importance is justified and the review aims are formulated. Key statements are supported by references. Appropriate evidence is present and relevant outcome data are presented accordingly.
Regrettably, literature search strategy is not described and inclusion criteria are not mentioned.
References do not appear in consecutive order.
The single table contains too much information. I would suggest to divide it in three tables, one for each pertinent section (sections 3, 4, and 5), to make it more readable.
Minor points:
Line 40: The sentence is not understandable. Something is missing: "The ESTRO and EORTC … oligometastatic state further distinguishes"
Line 60 and 86: Weickhardt et al. is reviewed two times with almost the same words.
Line 156: The study presented in Reference 21 included melanoma patients. It should be mentioned.
Lines 120 and 174: Both paragraphs are almost identical. The considerations formulated there are better fitted for section eight (Conclusions)
Author Response
I would like to congratulate the authors for a timely review of a hot oncologic topic in oncology. The article's importance is justified and the review aims are formulated. Key statements are supported by references. Appropriate evidence is present and relevant outcome data are presented accordingly.
Regrettably, literature search strategy is not described and inclusion criteria are not mentioned.
Thank you, the methods have now been updated to include the search strategy.
References do not appear in consecutive order.
We have corrected this. Thank you for bringing it to our attention.
The single table contains too much information. I would suggest to divide it in three tables, one for each pertinent section (sections 3, 4, and 5), to make it more readable.
Thank you for this comment. We considered doing this in the initial submission. The issue is that multiple studies represented in the table include both oncogenic driver positive and negative patients and do not necessarily fall in their silos of those oncogenic negative on a checkpoint inhibitor and those oncogenic positive on a tyrosine kinase inhibitor which is why we felt the best way to represent the limited number of studies currently is in one table.
Minor points:
Line 40: The sentence is not understandable. Something is missing: "The ESTRO and EORTC … oligometastatic state further distinguishes"
Thank you, this section has now been reworded.
Line 60 and 86: Weickhardt et al. is reviewed two times with almost the same words.
Thank you for bringing this to our attention. We have made the change.
Line 156: The study presented in Reference 21 included melanoma patients. It should be mentioned.
We have added this thank you.
Lines 120 and 174: Both paragraphs are almost identical. The considerations formulated there are better fitted for section eight (Conclusions)
Thank you we have adjusted this so that they are not identical and thank you for the suggestion.
Round 2
Reviewer 1 Report
Comments and Suggestions for Authors
The quality of the manuscript has been improved with addressing major issues/concerns.